# The burden of SARS-CoV-2 among healthcare workers across 16 hospitals of Kashmir, India—A seroepidemiological study

**Inaamul Haq[1], Mariya Amin Qurieshi[1]\*, Muhammad Salim Khan[1], Sabhiya Majid[2], Arif Akbar Bhat[2], Rafiya Kousar[1], Iqra Nisar Chowdri[1], Tanzeela Bashir Qazi[1], Abdul Aziz Lone[1], Iram Sabah[1], Misbah Ferooz Kawoosa[1], Shahroz Nabi[1], Ishtiyaq Ahmad Sumji[1], Shifana Ayoub[1], Mehvish Afzal Khan[1], Anjum Asma[1], Shaista Ismail[1]**

1 Department of Community Medicine, Government Medical College, Srinagar, Jammu and Kashmir, India,
2 Department of Biochemistry, Government Medical College, Srinagar, Jammu and Kashmir, India

\* maryaamin123@gmail.com

**Data Availability Statement:** All relevant data are within the manuscript and its Supporting Information files.

## Abstract

SARS-CoV-2 pandemic has greatly affected healthcare workers because of the high risk of getting infected. The present cross-sectional study measured SARS-CoV-2 antibody in healthcare workers of Kashmir, India. Methods: Serological testing to detect antibodies against nucleocapsid protein of SARS-CoV-2 was performed in 2003 healthcare workers who voluntarily participated in the study. Results: We report relatively high seropositivity of 26.8% (95% CI 24.8–28.8) for SARS-CoV-2in healthcare workers, nine months after the first case was detected in Kashmir. Most of the healthcare workers (71.7%) attributed infection to the workplace environment. Among healthcare workers who neither reported any prior symptom nor were they ever tested for infection by nasopharyngeal swab test, 25.5% were seropositive. Conclusion: We advocate interval testing by nasopharyngeal swab test of all healthcare workers regardless of symptoms to limit the transmission of infection within healthcare settings.

## Introduction

Healthcare workers (HCWs) represent a high-risk groupvulnerable to SARS-CoV-2 infection. They have been hard hit by the pandemic across the globe with the highest infectionreported from Europe and the USA [1]. The likelihood of becoming infected is manifold when compared to the general population [2]. The safety of HCWs is of utmost importance in the fight against the pandemic. With the introduction of the world's largest lockdown in history on 25th March 2020, India started the process of lifting restriction in phases and by September 2020 people were allowed to resume routine activity with compulsory use of face masks in public and workplace [3]. As of 17thJanuary2021, more than 10,571,773 cases and 152,419 deaths related to COVID-19 were reported in the country. Among them, 72,491 cases and 1209 deaths were from Kashmir [4, 5]. More than 1000 HCWs died due to SARS-CoV-2 infection in India. In Kashmir, around 420 HCWs were reportedly infected by the end of August 2020 [6].

**Funding:** Initials of the authors who received each award: SMSK. • Grant numbers awarded to each author: Order No: 50 of 2020 Dated 3-9-2020 • The full name of each funder: National Health Mission, Jammu and Kashmir. The funders had no role in study design, data collection and analysis, decision to publish, or preparation of the manuscript.

**Competing interests:** The authors have declared that no competing interest exist.

Seroprevalence studies are a well-established approach to determine the burden of SARS-CoV-2. They help in understanding the extent of infection as they are used as a proxy measure for previous infections. In settings like hospitals, they give clues about the dynamics of transmission of infection within a closed situation. Such information helps in policymaking and evaluating the intervention from time to time [7, 8]. We conducted this study towards the end of the first wave of the pandemic in the country. The study was done to determine the serological status of HCWs, irrespective of prior SARS-CoV-2 infection, in hospitals of Kashmir, India.

## Material and methods

### Study design and settings

We conducted a cross-sectional study across sixteen hospitals (Level II and Level III) of the Kashmir Division. Level II hospitals are equipped to provide comprehensive secondary care including specialist services to the local populace. They play a pivotal role in epidemics and disaster management. Level III hospitals are referral hospitals that provide super specialty care and have an additional distinction of being training and teaching institutions. Kashmir Division has ten districts (administrative unit for governance), each having a Level II hospital-District Hospital. In addition, the central district which is also the capital of Kashmir Division has eight tertiary care hospitals (Level III). In response to the pandemic, all the Level III hospitals across Kashmir were equipped with intensive care units, ventilators, and beds with reliable oxygen supplies. The Level II hospitals were provided ventilators and beds with reliable oxygen supplies based on the expected patient load. HCWs of all hospitals were rigorously trained on Infection Prevention and Control (IPC) and the use of Personal Protective Equipment.

### Study population and data collection

We sought permission from the administrative heads of all Level II and Level III hospitals of Kashmir. Of the 18 hospitals, two tertiary care hospitals did not agree to participate. The HCWs of hospitals were communicated about the nature of the study through their administrative heads and were offered voluntary testing for antibodies against SARS-CoV-2 from 14th to 17th January 2021. Prior information was sent about the days of visit through a liaison person identified at each hospital. A designated team then visited these hospitals to collect information on sociodemographic variables, history of past infection, symptoms, and nasopharyngeal swab testing for SARS-CoV-2 from the consenting HCWs. Among HCWs who reported SARS-CoV-2 infection since the beginning of the pandemic, we specifically inquired about their attribution of infection- healthcare setting or community. We collected information on an interview form generated in a user-friendly and free mobile application [9]. Following the interview, a venous blood sample was collected which was packed and transported under the protocol to the designated laboratory with a testing facility for antibodies against SARS-CoV-2.

Sample size and sample selection: We did not calculate a priori sample size. The participation of HCWs in the study was voluntary. Sample selection was not carried out due to time constraints and practical difficulty.

### Laboratory procedure

We used chemiluminescent microparticle immunoassay for the qualitative detection of IgG against SARS-CoV-2 nucleocapsid protein. The manufacturer's reported sensitivity and specificity for the test are 100% and 99% respectively [10]. The test measures the number of IgG

antibodies to SARS-CoV-2 in the serum sample in relative light units. The IgG threshold value of ≥1.4 was considered positive, which conforms with the manufacturer's instructions.

## Statistical analysis

Our primary outcome was to estimate the seroprevalence of SARS-CoV-2 in HCWs of Kashmir. We estimated the seroprevalence of SARS-CoV-2 as the proportion of HCWs with a positive result on immunoassay. We adjusted the seroprevalence for sensitivity and specificity of the test with the help of the formula: Adjusted seroprevalence = (Unadjusted seroprevalence + Specificity −1) ÷ (Sensitivity + Specificity −1) [11]. We used the chi-square test to compare seroprevalence by age, gender, work category, past symptoms, and nasopharyngeal test result. Statistical significance was defined at $p < 0.05$. We analyzed data using Stata, version 15.1 (StataCorp LP).

## Ethics

The Institutional Ethical Committee approved the study and we obtained written informed consent from all participants. Study participation was voluntary.

## Results

A total of 2003 HCWs which include 752 females and 1251 males, from 16 hospitals of Kashmir voluntarily participated in the study. Hospital-wise details of participation of HCWs are provide in S1 Table. The mean age was 37.4 years (10.7 SD). The participants included doctors (28.1%), medical technicians (23.5%), nurses (18.6%), housekeeping personnel (19.3%), and administrative staff (10.5%) (Table 1).

Overall adjusted seroprevalence was 26.8% (95% CI 24.8–28.8). The adjusted seroprevalence across health facilities of ten district hospitals is shown in Fig 1.

Fig 1 shows the district-wise cumulative number of reported cases of SARS-CoV-2 infection and seroprevalence among HCWs of the hospitals within these districts, two weeks before the study (31st Dec 2021). We looked for the correlation between the caseload in the community and seroprevalence in our study. We did not find a strong correlation between the number of reported cases in a district and seroprevalence (r = 0.410).

Gender and the occupational group were not statistically related to seropositivity. Age >40 years, having symptoms in the past, nasopharyngeal swab test positivity was significantly associated with seroprevalence ($p < 0.05$).

Of those who ever had SARS-CoV-2 infection since the beginning of the pandemic, 71.7% (157/219) attributed it to the workplace, 17.3% (38/219) had community exposure either from family, friend or neighbourhood and 11% (24/219) were not sure about possible place of exposure.

A total of 985 HCWs had undergone nasopharyngeal swab testing, of whom 219 (22.2%) self-reported a positive test. Among them, 150/219, 66.5% were seropositive. A sizeable proportion (25.5% (249/978)) of HCWs who had never tested for SARS-CoV-2 infection nor reported any symptom compatible with infection tested positive for IgG. On the contrary, of 276 HCWs who had ever tested for SARS-CoV-2 infection and were symptomatic, only 49.6% (137) were seropositive. Of the 166 HCWs who self-reported positive nasopharyngeal swab test and had symptoms, 32.5% (54/166) were seronegative (Table 2).

We also analyzed the reported time since nasopharyngeal swab positivity. Among those who reported positive nasopharyngeal swab test at least six months before, only 41.7% (10/24) were currently seropositive. The corresponding figures were 64.4% (93/144) and 68.1%

**Table 1. Seroprevalence of SARS-CoV-2 IgG antibodies by sociodemographic features of health care workers.**

| | | Number of participants | Test adjusted Seroprevalence % (95% CI) | *P* value |
|---|---|---|---|---|
| Overall Seroprevalence | | 2003 | 26.8 (24.8–28.8) | |
| Gender | Male | 1251 | 25.2 (22.8–27.7) | 0.669 |
| | Female | 752 | 26.1 (23.0–29.4) | |
| Age | ≤ 40 | 1330 | 23.2 (20.9–25.6) | 0.001 |
| | >40 | 673 | 30.1 (26.7–23.7) | |
| Designation | Doctor | 562 | 25.6 (22.0–29.4) | 0.937 |
| | Nursing staff | 373 | 24.8 (20.6–29.5) | |
| | Medical technician | 470 | 26.5 (22.6–30.8) | |
| | Housekeeping | 387 | 24.6 (20.4–29.3) | |
| | Administrative staff | 211 | 26.0 (20.3–32.4) | |
| Past symptoms for SARS-CoV-2 infection | Yes | 316 | 46.4 (40.9–51.9) | 0.0001 |
| | No | 1687 | 21.6 (19.6–23.7) | |
| NPS test result | Positive | 219 | 66.5 (60.0–72.3) | 0.0001 |
| | Negative | 766 | 15.8 (13.2–18.6) | |

CI- confidence Interval.

NPS- Nasopharyngeal Swab.

(143/210) among those who reported positive NPS test three months and one month back respectively (Fig 2).

Fig 3 shows the mean difference in IgG index value, 30 days, 90 days, and 180 days after initial NPS test positivity across age, gender and symptoms.

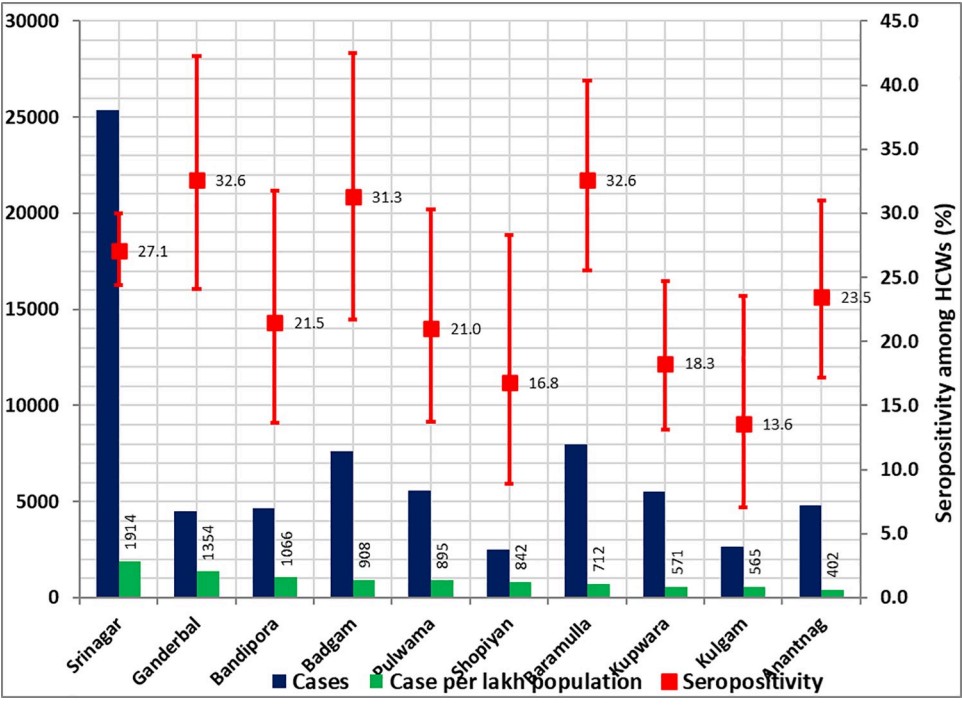

**Fig 1. District–wise SARS-CoV-2 seroprevalence (95% CI) in healthcare workers of Kashmir and reported cases of infection in the population.**

**Table 2. Nasopharyngeal swab test characteristics, symptomatology, and seropositivity for SARS-CoV-2 specific IgG.**

| | | No. of participants | Seropositive n(%) | Seronegative n (%) |
|---|---|---|---|---|
| Ever tested for SARS-CoV-2 | Symptoms | | | |
| Yes | Present | 276 | 137 (49.6) | 139 (50.4) |
| | Absent | 709 | 149 (21) | 560 (79) |
| No | Present | 40 | 16 (40) | 24 (60) |
| | Absent | 978 | 249 (25.5) | 728 (74.5) |
| Nasopharyngeal swab Test result | | | | |
| Positive | Present | 166 | 112 (67.5) | 54 (32.5) |
| | Absent | 53 | 38 (71.7) | 15 (28.3) |
| Negative | Present | 110 | 25 (22.7) | 85 (77.3) |
| | Absent | 656 | 111 (16.9) | 545 (83.1) |

## Discussion

Serological testing is considered complementary to the nasopharyngeal swab testing in the detection of the true burden of SARS-CoV-2 infection. It provides relevant information in understanding the transmission dynamics within specific population groups.

The present serosurvey was conducted at a time when the country was reporting a decline in daily new cases of SARS-CoV-2. To sustain the progress made towards containment of the spread of SARS-CoV-2 infection, like other countries, India also began preparing for the introduction of the vaccine in the country focussing on HCWs in the initial phase [11, 12].

We report the serological status of 2003 HCWs from Level II and Level III hospitals of Kashmir which is the northernmost territory of India. The seroprevalence of SARS-CoV-2

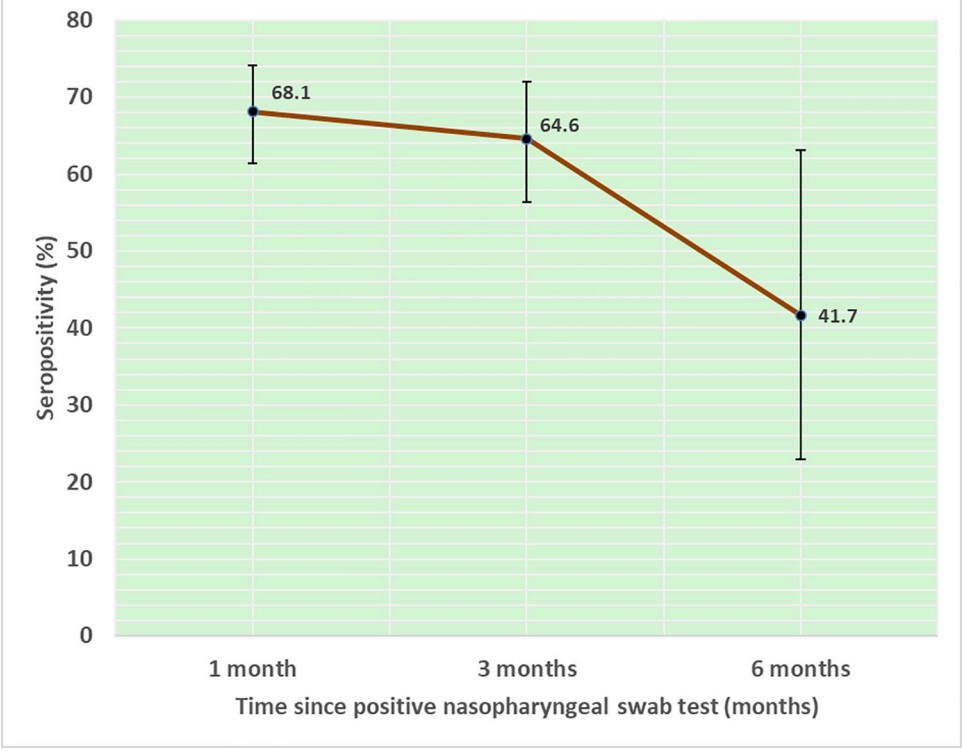

**Fig 2. SARS-CoV-2 seropositivity with 95% CI and time (months) since nasopharyngeal swab test.**

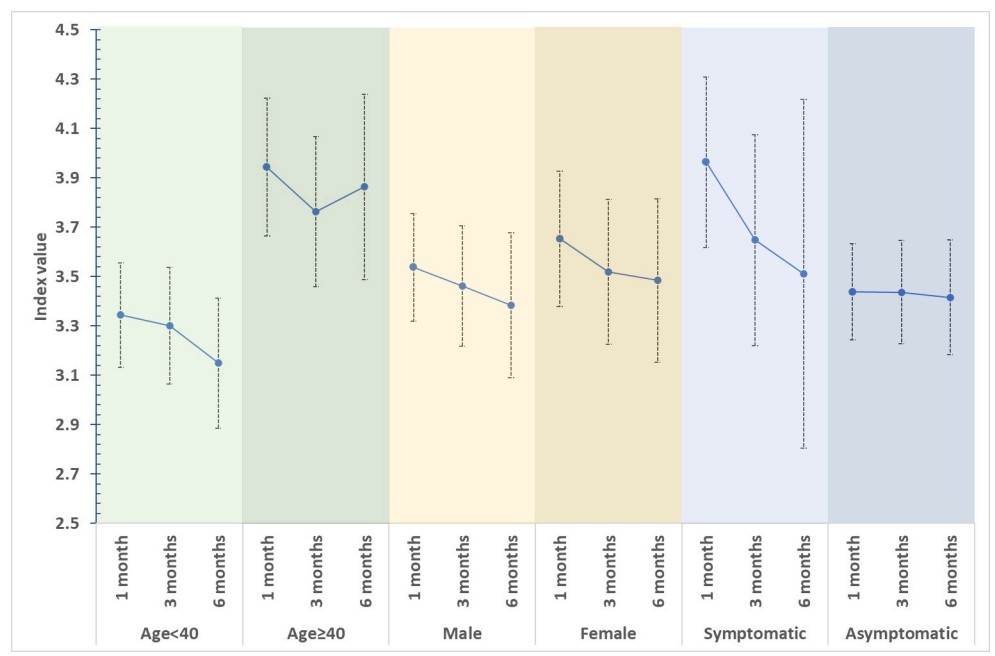

**Fig 3. Mean difference in IgG index value, 30 days, 90 days, and 180 days after initial NPS test positivity across age, gender and symptoms.**

adjusted for test characteristics was 26.8% (95% CI 25.6–29.5) in HCWs across 16 different hospitals, ten months after the first case was reported from Kashmir, India. A similar study was conducted in June 2020 in the capital city of Srinagar [13]. As compared to June 2020, in January 2021, there is an eleven-fold increase in the seroprevalence in HCWs. The findings of our study are in concordance with the recent nationwide serosurvey in India which showed 25.7% HCWs in the country had serological evidence of SARS-CoV-2 infection by the end of December 2020 [14]. However, there is a huge variation in the reported seroprevalence of SARS-CoV-2 in HCWs across the globe in the last year ranging from 0.34% in Tokyo to 13.7% in New York hospitals [15, 16]. The variation may be attributed to the difference in the extent of infection in the population, the timing of data collection, the quality of the study, and the nature of testing. Despite these reasons, in certain settings like in the USA and Europe who were worst affected by this pandemic, the seropositivity in the HCWs is quite less as compared to India [8, 17]. This could be explained by better infection prevention and control practices and adherence to protocol in their hospitals. With just above 1% GDP spent on public health in India, which is lower than in neighbouring low-income countries, the existing health infrastructure was not able to cope up with the increase in demand for health care services, resulting in an acute shortage of PPE and thus exposing HCWs to infection [18]. Lack of discipline and non-compliance to adherence to COVID appropriate behaviour among HCWs in and outside healthcare settings is another possible reason for higher seropositivity in the study population.

We observed an age differential in seropositivity with higher seroprevalence in the age group greater than 40 years. In this regard, there are varied reports from different hospital settings [19–22]. Although there is no strong evidence yet to suggest that the age of an individual determines the likelihood of acquiring SARS-CoV-2 infection, impaired immunity with increasing age is a known age-related phenomenon [23–25]. This increases the susceptibility to various infections including SARS-CoV-2. As the research on SARS-CoV-2 infection is still emerging this aspect needs to be further explored.

We did not observe any significant difference in seropositivity among categories of HCWs. This possibly reflects the extent of transmission from HCW to HCW within the healthcare environment. Nearly three-fourths (71.7%) of HCWs who self-reported SARS-CoV-2 infection attributed it to the workplace. Similar findings are reported by seroprevalence studies on HCWs done towards the end of last year [14, 26].

We found 23.6% seropositivity among asymptomatic HCWs and 17.8% of seropositive HCW were not previously diagnosed with SARS-CoV-2 infection when tested by NPS. It is important to know that 25.5% of seropositive HCWs were never tested for infection. Similar results are reported by other studies done on HCWs [27, 28]. Asymptomatic individuals also labeled as super spreaders of infection pose a tremendous challenge in the containment of transmission of SARS-CoV-2 infection in the present pandemic in all settings [29]. They pose a serious risk for their fellow workers, the patient population as well as their families and community. This highlights the need for strict surveillance of HCWs by performing interval testing from time to time by RT-PCR, irrespective of having symptoms to ensure the safety of other HCWs [30, 31].

To get an insight into the period for which antibodies remain detectable in previously SARS-CoV-2 positive individuals, we asked about the time since they have had their NPS test positive. We observed 143 out of 210 (68%) had detectable antibodies one month after infection as against 10 out of 24 (42%) at six months.

We observed a decline in mean IgG index value across time with a higher mean index value of IgG for age >40 years, males, and those who reported symptoms. More importantly, nine of 10 who self-reported symptoms retained seropositivity six months after testing positive for infection. There is a paucity of data on the longitudinal assessment of serological assays. So far the only large-scale data with a very long follow-up period of assessment of the sustainability of antibodies following infection with SARS-CoV-2 has been reported from the United States using laboratory data. They reported high seropositivity (87.5%) 300 days after infection. The longevity of seropositivity exhibited significant relation with age and sex with quicker decay in the old age group and males [32]. However, they did not look for differences based on symptoms and their analysis was based on a qualitative assessment of serology.

Since we lack information on the initial seroconversion, we are not sure whether current seronegativity is due to a decline in the antibody level or failure to mount antibody response in the first place. Keeping in view the resource constraints, a longitudinal study would be difficult to conduct in our setting to measure the duration for seropositivity. Further seropositivity does not guarantee protection against reinfection so it is imperative to highlight the waning seropositivity and use it as an argument to initiate vaccination for the protection of population beginning with high-risk groups.

Countries across the globe have already started the vaccination campaign against SARS-CoV-2 infection. However, it is faced with multiple challenges ranging from the effectiveness of the vaccine in a real-world setting, breakthrough infections among vaccinated people, and the emergence of virus variants. Robust surveillance and epidemiological studies are further required to understand the complex dynamics of SARS-CoV-2 infection in post vaccination period.

## Strengths and limitations

We used a fairly accurate, high throughput serological test for the detection of IgG against SARS-CoV-2Since we included hospitals across all districts in Kashmir, the findings can therefore be considered representative for the HCWs in Kashmir.

Our study has certain limitations. We did not evaluate the validity of the test kits in-house. However, we adjusted seroprevalence estimates for test performance characteristics. We used

a qualitative assessment of IgG antibodies against SARS-CoV-2 nucleocapsid protein alone. We did not test for antibodies against the spike protein. Due to resource constraints, we could not assess neutralizing antibodies or IgM antibodies. Owing to the self-selected nature of participants the seroprevalence could be overestimated and the possibility of selection bias cannot be ruled out. However, we trust that it does not have much influence on the study validity owing to the similar working environment of responders and non-responders.

## Conclusion

We report relatively high seropositivity for SARS-CoV-2 in HCWs, nine months after the first case was detected in Kashmir. High seropositivity in asymptomatic HCWs should be taken seriously as they remain unaccounted for. They act as silent spreaders of infection which can further incapacitate the strained healthcare system in this pandemic. We advocate the HCWs should undergo interval testing by nasopharyngeal swab test for timely detection of infection regardless of symptoms to limit the in-hospital transmission of infection among HCWs and from infected HCWs to patients.

## Supporting information

**S1 Table. Hospital-wise details of participation of HCWs across 16 hospitals.**
(DOCX)

**S1 Data.**
(DTA)

## Acknowledgments

The authors appreciate the support extended by Prof. Samia Rashid, Principal, Government Medical College Srinagar; Department of Biochemistry, Government Medical College Srinagar; Directorate of Health Services, Kashmir; and Chief Medical Officers of all District hospitals across Kashmir Division. We thank the medical interns involved in this study.

## Author Contributions

**Conceptualization:** Inaamul Haq, Mariya Amin Qurieshi, Muhammad Salim Khan.

**Data curation:** Inaamul Haq, Mariya Amin Qurieshi, Arif Akbar Bhat, Rafiya Kousar, Iqra Nisar Chowdri, Tanzeela Bashir Qazi, Abdul Aziz Lone, Iram Sabah, Misbah Ferooz Kawoosa, Shahroz Nabi, Ishtiyaq Ahmad Sumji, Shifana Ayoub, Mehvish Afzal Khan, Anjum Asma, Shaista Ismail.

**Formal analysis:** Inaamul Haq, Mariya Amin Qurieshi.

**Funding acquisition:** Muhammad Salim Khan, Sabhiya Majid.

**Investigation:** Inaamul Haq, Mariya Amin Qurieshi.

**Methodology:** Inaamul Haq, Mariya Amin Qurieshi, Muhammad Salim Khan.

**Project administration:** Muhammad Salim Khan, Sabhiya Majid.

**Resources:** Muhammad Salim Khan, Arif Akbar Bhat, Rafiya Kousar, Iqra Nisar Chowdri, Tanzeela Bashir Qazi, Abdul Aziz Lone, Iram Sabah, Misbah Ferooz Kawoosa, Shahroz Nabi, Ishtiyaq Ahmad Sumji, Shifana Ayoub, Mehvish Afzal Khan, Anjum Asma, Shaista Ismail.

**Supervision:** Inaamul Haq, Mariya Amin Qurieshi, Muhammad Salim Khan, Arif Akbar Bhat, Rafiya Kousar, Iqra Nisar Chowdri, Tanzeela Bashir Qazi, Abdul Aziz Lone, Iram Sabah, Misbah Ferooz Kawoosa, Shahroz Nabi, Ishtiyaq Ahmad Sumji, Shifana Ayoub, Mehvish Afzal Khan, Shaista Ismail.

**Validation:** Muhammad Salim Khan, Sabhiya Majid, Arif Akbar Bhat, Anjum Asma.

**Visualization:** Sabhiya Majid, Arif Akbar Bhat.

**Writing – original draft:** Inaamul Haq, Mariya Amin Qurieshi.

**Writing – review & editing:** Inaamul Haq, Mariya Amin Qurieshi, Muhammad Salim Khan, Sabhiya Majid, Arif Akbar Bhat, Rafiya Kousar, Iqra Nisar Chowdri, Tanzeela Bashir Qazi, Abdul Aziz Lone, Iram Sabah, Misbah Ferooz Kawoosa, Shahroz Nabi, Ishtiyaq Ahmad Sumji, Shifana Ayoub, Mehvish Afzal Khan, Anjum Asma, Shaista Ismail.

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
