## [Decision Letter · Decision Letter 0]

5 Aug 2021

PONE-D-21-21086

The burden of SARS-CoV-2 among healthcare workers across 16 hospitals of Kashmir, India- a seroepidemiological study

PLOS ONE

Dear Dr. Mariya Amin Qureishi,

Thank you for submitting your manuscript to PLOS ONE. After careful consideration, we feel that it has merit but does not fully meet PLOS ONE’s publication criteria as it currently stands. Therefore, we invite you to submit a revised version of the manuscript that addresses the points raised during the review process.

Please review the comments made by the reviewers and provide point by point response in the revised manuscript.

We look forward to receiving your revised manuscript.

Kind regards,

Muhammad Adrish, MD, MBA, FCCP, FCCM

Academic Editor

PLOS ONE

Journal Requirements:

Reviewers' comments:

Reviewer's Responses to Questions

**Comments to the Author**

1. Is the manuscript technically sound, and do the data support the conclusions?

Reviewer #1: Yes

Reviewer #2: Partly

2. Has the statistical analysis been performed appropriately and rigorously? 

Reviewer #1: Yes

Reviewer #2: Yes

3. Have the authors made all data underlying the findings in their manuscript fully available?

Reviewer #1: Yes

Reviewer #2: No

4. Is the manuscript presented in an intelligible fashion and written in standard English?

Reviewer #1: Yes

Reviewer #2: Yes

5. Review Comments to the Author

Reviewer #1: This is an important study on a relevant and timey topic. The data would be useful, although we are now past the second wave in formulatng public health policy. I would suggest adding a note on the implications of vaccination status in future studies or longitudinal studies on seroprevalence

Reviewer #2: This study is a cross sectional estimate of COVID seroprevalence based on the nucleocapsid IgG antibody test.

They invited all the level 2 and 3 hospitals in Kashmir. 2 of them refused invitation. From those who agreed to participate, they invited willing hospital staff to enroll.

This serology survey was conducted in January 14 to 17, at a time when first wave had ceased, and Vaccines were yet to be started.

- What is the total number of HCW in each of the hospitals. What % of the staff population accepted to take part in the study?

- why sample size was not calculated?

- Why randomisation was not done?

- What is the mean age of HCWs overall - those included + those refused consent?

- What was the PCR positivity prevalence in those who did not take part?

- Comorbidities have not been captured!

- Was appropriate PPE available for all staff at all times? Did Housekeeping staff get hazmut? A note on the level of preparedness for COVID19 among HCW in Kashmir should be added to help the reader better understand the background.

- Are administrative staff at increased risk of exposure to Clinical areas? Did all the included HCWs work in COVID wards or Emergency services?

- Severity of COVID infection among those who were positive and the Antibody trend at 1,3 and 6 months could have been captured

Overall the study has good data but needs to be presented in a clear way.

6. PLOS authors have the option to publish the peer review history of their article (what does this mean?). If published, this will include your full peer review and any attached files.

Reviewer #1: **Yes: **Aneesh Basheer

Reviewer #2: No

---

## [Author Response · Author response to Decision Letter 0]

21 Aug 2021

Reviewer #1: This is an important study on a relevant and timely topic. The data would be useful, although we are now past the second wave in formulating public health policy. I would suggest adding a note on the implications of vaccination status in future studies or longitudinal studies on seroprevalence.

Response: Thank you for your valuable suggestion. It has now been added in the discussion section of the manuscript.

Reviewer #2: This study is a cross-sectional estimate of COVID seroprevalence based on the nucleocapsid IgG antibody test. They invited all the level 2 and 3 hospitals in Kashmir. 2 of them refused the invitation. From those who agreed to participate, they invited willing hospital staff to enroll. This serology survey was conducted from January 14 to 17, at a time when the first wave had ceased, and Vaccines were yet to be started.

What is the total number of HCW in each of the hospitals? What % of the staff population accepted to take part in the study?

Response: Breakup of HCWs in each hospital is tabulated. Overall 40.8% HCWs took part in this study. We have added this information in the manuscript as a supplementary file

Hospital No. of HCWs No. of HCWs participated

District hospital Ganderbal 195 104

District hospital Bandipora 108 85

District hospital Badgam 110 78

District hospital Pulwama 307 100

District hospital Shopiyan 73 64

District hospital Baramulla 174 159

District hospital Kupwara 200 187

District hospital Kulgam 214 77

District hospital Anantnag 206 153

JLNM hospital Srinagar 132 120

SMHS & SS hospital Srinagar 2200 244

Chest disease hospital Srinagar 210 43

Maternity hospital Srinagar 255 142

Bone and Joint hospital Srinagar 215 184

Pediatric hospital Srinagar 217 180

IMHANS hospital Srinagar

 95 83

Total 4911 2003 (40.8%)

Why randomisation was not done?

Response: This was a cross-sectional study, so randomization does not apply here. I believe the reviewer is asking about random sampling here. We did not perform a random sampling strategy in this study due to time constraints and practical difficulty. 

why the sample size was not calculated?

Response: Our results are based on the voluntary participation of HCWs. Our study included 2003 HCWs and the seroprevalence estimates of 26.8 % with a confidence interval of (24.8- 28.8) show that the estimates have pretty good precision. 

- What is the mean age of HCWs overall - that refused consent?

Response: Thank you for your query. The information of those who did not participate in the study was not collected. We have included it in the limitation section of the manuscript.

What was the PCR positivity prevalence in those who did not take part?

Response: Thank you for your query. We did not collect the information of those who did not participate in the study.

Comorbidities have not been captured!

Response: Information about comorbidities was not collected

- Was appropriate PPE available for all staff at all times? Did the Housekeeping staff get hazmat? A note on the level of preparedness for COVID19 among HCW in Kashmir should be added to help the reader better understand the background.

Response: Thank you. Owing to resource constraints, the PPE was not available for all staff at all times. Hazmut was not provided to the housekeeping staff. A note on the level of preparedness has now been added to the methods section.

 Our administrative staff at increased risk of exposure to Clinical areas? Did all the include HCWs work in COVID wards or Emergency services?

Response: Since the study was done 10 months after the appearance of the first case of SARS-CoV-2 infection in Kashmir, the level of exposure among all categories of healthcare workers at this time was not much different. All the hospitals had restarted their routine outpatient and inpatient activities. The same is reflected in our study as well. We did not find any difference in the seroprevalence among different occupational categories of HCWs.

---

## [Decision Letter · Decision Letter 1]

27 Sep 2021

PONE-D-21-21086R1The burden of SARS-CoV-2 among healthcare workers across 16 hospitals of Kashmir, India- a seroepidemiological studyPLOS ONE

Dear Dr. Qurieshi,

Thank you for submitting your manuscript to PLOS ONE. After careful consideration, we feel that it has merit but does not fully meet PLOS ONE’s publication criteria as it currently stands. Therefore, we invite you to submit a revised version of the manuscript that addresses the points raised during the review process.

ACADEMIC EDITOR: Please review comments made by reviewers and provide response in your revised manuscript.

We look forward to receiving your revised manuscript.

Kind regards,

Muhammad Adrish, MD, MBA, FCCP, FCCM

Academic Editor

PLOS ONE

Journal Requirements:

Reviewers' comments:

Reviewer's Responses to Questions

**Comments to the Author**

1. If the authors have adequately addressed your comments raised in a previous round of review and you feel that this manuscript is now acceptable for publication, you may indicate that here to bypass the “Comments to the Author” section, enter your conflict of interest statement in the “Confidential to Editor” section, and submit your "Accept" recommendation.

Reviewer #1: All comments have been addressed

Reviewer #2: All comments have been addressed

2. Is the manuscript technically sound, and do the data support the conclusions?

Reviewer #1: Yes

Reviewer #2: Partly

3. Has the statistical analysis been performed appropriately and rigorously? 

Reviewer #1: Yes

Reviewer #2: Yes

4. Have the authors made all data underlying the findings in their manuscript fully available?

Reviewer #1: Yes

Reviewer #2: Yes

5. Is the manuscript presented in an intelligible fashion and written in standard English?

Reviewer #1: Yes

Reviewer #2: Yes

6. Review Comments to the Author

Reviewer #1: In the first review, implications of vaccination was sought. The authors have repsonded by adding a section on this in the discussion. This appears satisfactory and reflects the uncertainty regarding the effect of vaccine induced antibody responses on the seroprevalence among healthcare workers. The paper can be accepted.

Reviewer #2: This study is a cross-sectional estimate of COVID seroprevalence based on the nucleocapsid IgG antibody test. They invited all the level 2 and 3 hospitals in Kashmir. 2 of them refused the invitation. From those who agreed to participate, they

invited willing hospital staff to enroll. This serology survey was conducted from January 14 to 17, at a time when the first wave had ceased, and Vaccines were yet to be started.

Kindly add the reasons provided by the authors for not calculating sample size and convenience sampling in the methodology section.

7. PLOS authors have the option to publish the peer review history of their article (what does this mean?). If published, this will include your full peer review and any attached files.

Reviewer #1: **Yes: **Aneesh Basheer

Reviewer #2: No

---

## [Author Response · Author response to Decision Letter 1]

12 Oct 2021

Reviewer #2: This study is a cross-sectional estimate of COVID seroprevalence based on the nucleocapsid IgG antibody test. They invited all the level 2 and 3 hospitals in Kashmir. 2 of them refused the invitation. From those who agreed to participate, they invited willing hospital staff to enroll. This serology survey was conducted from January 14 to 17, at a time when the first wave had ceased, and Vaccines were yet to be started.

Kindly add the reasons provided by the authors for not calculating sample size and convenience sampling in the methodology section.

Response: Thank you for your suggestion. We have now included it in the methodology section of the manuscript.

---

## [Decision Letter · Decision Letter 2]

29 Oct 2021

The burden of SARS-CoV-2 among healthcare workers across 16 hospitals of Kashmir, India- a seroepidemiological study

PONE-D-21-21086R2

Dear Dr. Qurieshi,

We’re pleased to inform you that your manuscript has been judged scientifically suitable for publication and will be formally accepted for publication once it meets all outstanding technical requirements.

Kind regards,

Laith Al-Eitan

Academic Editor

PLOS ONE

Additional Editor Comments (optional):

Thanks for addressing all comments raised by reviewers

Reviewers' comments:

Reviewer's Responses to Questions

**Comments to the Author**

1. If the authors have adequately addressed your comments raised in a previous round of review and you feel that this manuscript is now acceptable for publication, you may indicate that here to bypass the “Comments to the Author” section, enter your conflict of interest statement in the “Confidential to Editor” section, and submit your "Accept" recommendation.

Reviewer #1: All comments have been addressed

Reviewer #2: All comments have been addressed

2. Is the manuscript technically sound, and do the data support the conclusions?

Reviewer #1: Yes

Reviewer #2: Yes

3. Has the statistical analysis been performed appropriately and rigorously? 

Reviewer #1: Yes

Reviewer #2: Yes

4. Have the authors made all data underlying the findings in their manuscript fully available?

Reviewer #1: Yes

Reviewer #2: Yes

5. Is the manuscript presented in an intelligible fashion and written in standard English?

Reviewer #1: Yes

Reviewer #2: Yes

6. Review Comments to the Author

Reviewer #1: No revisions were suggested by this reviewer in the previous version. The authors have justified their reason for not calculating sample size raised by the second reviewer.

Reviewer #2: The manuscript in its current form appears to be refined and acceptable for Publication. No further queries from my end.

7. PLOS authors have the option to publish the peer review history of their article (what does this mean?). If published, this will include your full peer review and any attached files.

Reviewer #1: **Yes: **Aneesh Basheer

Reviewer #2: No

---

## [Editor Report · Acceptance letter]

10 Nov 2021

PONE-D-21-21086R2 

The burden of SARS-CoV-2 among healthcare workers across 16 hospitals of Kashmir, India- a seroepidemiological study 

Dear Dr. Qurieshi:

I'm pleased to inform you that your manuscript has been deemed suitable for publication in PLOS ONE. Congratulations! Your manuscript is now with our production department. 

Kind regards, 

on behalf of

Dr. Laith Al-Eitan 

Academic Editor

PLOS ONE